# Home Artificial Nutrition and Energy Balance in Cancer Patients: Nutritional and Clinical Outcomes

**DOI:** 10.3390/nu14204307

**Published:** 2022-10-14

**Authors:** Enrico Ruggeri, Rita Ostan, Silvia Varani, Raffaella Pannuti, Guido Biasco

**Affiliations:** 1Training and Research Department, National Tumor Assistance (ANT) Foundation, Via Jacopo di Paolo, 36-40128 Bologna, Italy; 2Department of Experimental, Diagnostic and Specialty Medicine, University of Bologna, 40138 Bologna, Italy

**Keywords:** home artificial nutrition, cachexia, malnutrition, energy balance, cancer, palliative care

## Abstract

Malnutrition is one of the main factors determining cachexia syndrome, which negatively impacts the quality of life and survival. In cancer patients, artificial nutrition is considered as an appropriate therapy when the impossibility of an adequate oral intake worsened nutritional and clinical conditions. This study aims to verify, in a home palliative care setting for cancer patients, if home artificial nutrition (HAN) supplies a patient’s energy requirement, improving nutritional and performance status. A nutritional service team performed counseling at a patient’s home and assessed nutritional status (body mass index, weight loss in the past 6 months), resting energy expenditure (REE), and oral food intake; Karnofsky Performance Status (KPS); cachexia degree; and survival. From 1990 to 2021, 1063 patients started HAN. Among these patients, 101 suspended artificial nutrition for oral refeeding. Among the 962 patients continuing HAN until death, 226 patients (23.5%) survived 6 weeks or less. HAN allowed to achieve a positive energy balance in 736 patients who survived more than 6 weeks, improving body weight and KPS when evaluated after 1 month of HAN. Advanced cancer and cachexia degree at the entry of the study negatively affected the positive impact of HAN.

## 1. Introduction

Cachexia is among the main causes of morbidity and mortality in oncology [1,2], affecting more than 50% of patients with cancer and up to 80% of patients with head, neck, and gastrointestinal tumors [3]. Cachexia is the cause for up to 20% of cancer deaths [4]. Malnutrition represents one of the factors determining the degree of cachexia, defined as a “wasting” disorder characterized by weight loss, sarcopenia, hypercatabolism, and cancer systemic inflammation [5,6]. In cancer patients, the weight loss is due to both an inadequate caloric intake [7] and an increased energy expenditure due to cancer [8,9].

Artificial nutrition is an appropriate nutritional support in these patients, even in advanced stages of the disease [10], when the impossibility of an adequate oral intake leads to a worsening of clinical and nutritional conditions, performance status, and the quality of life [11]. To maintain nutritional status, artificial nutrition must provide a protein-caloric intake adequate to make positive the energy balance of the cancer patient [12]. The impact of artificial nutrition on the quality of life must always be considered. ESPEN guidelines for nutrition in cancer patients pointed out that artificial nutrition can be supplied within a palliative care program when a benefit for the quality of life is foreseeable and the risk of dying from malnutrition exceeds the possibility to die from cancer progression [12]. A recent study showed that weight loss was related to a significant impairment of the quality of life, highlighting the importance of nutritional interventions able to preserve or improve the nutritional status of patients with incurable cancer [13].

The home is one of the main settings where advanced cancer patients are assisted by palliative care programs. The presence of a specialized nutritional team offers the opportunity to develop a long-term program of artificial nutrition at home, satisfying the nutritional needs of the patients without causing additional burdens and distress to the patient and family [14]. The selection of patients for home artificial nutrition (HAN) becomes a critical decision for the nutritionist, who needs valid and proper eligibility criteria [11] to be sure that the benefits of artificial nutrition are superior than the possible harm, minimizing the hazard of its improper and excessive use.

The aim of this retrospective observational study was to verify, in a home palliative care setting for cancer patients, if HAN was a feasible and useful procedure to supply patient’s energy requirement, improving nutritional and performance status.

## 2. Materials and Methods

### 2.1. Nutritional Counseling

A multidisciplinary team of doctors, nurses, and psychologists, all of whom are trained in palliative care, provides around-the-clock 24-hour/7-day assistance to cancer patients as part of the home care model run by the National Assistance Tumor (ANT) Foundation [15]. 

The home palliative care physician requested the nutritional counseling for patients with malnutrition or progressive weight loss, who had a Karnofsky Performance Status (KPS [16]) of 40 or higher and not in the end of life (life expectancy more than 6 weeks, evaluated by Palliative Prognostic Score [17]).

The nutritional service team (NST), consisting of a gastroenterologist-nutritionist and a nutrition-dedicated nurse, has been working in ANT since 1990. 

The nutritionist evaluated clinical and nutritional status and stated the patient’s eligibility to HAN by completing four consecutive steps, as shown in Table 1 (for details see [11,18]).

The NST set up artificial nutrition at home in the majority of cases (62%). However, when artificial nutrition was set up in the hospital (38% of the cases), the ANT nutritionist provided counseling once the patient returned home and decided whether or not to continue nutritional therapy.

### 2.2. Pathogenesis of Malnutrition

The decision to start HAN stems from a careful examination of the pathogenesis of malnutrition and/or inadequate nutrient intake [11,19]. The primary condition was cancer-related organic consequences (dysphagia or intestinal obstruction) or therapeutic interventions (surgery or radio-chemotherapy). When hypermetabolism or anorexia was the cause of malnutrition, pharmacologic therapy (anticytokine agents, anabolic agents, metabolism inhibitors, or appetite stimulants) was tried [20]. If drug therapy failed, there was the indication to initiate HAN.

### 2.3. Evaluation of the Degree of Cachexia

The nutritionist evaluated cachexia and classified its stage (pre-cachexia, cachexia, or refractory cachexia) based on nutritional status (BMI and weight loss), presence of anorexia and sarcopenia, functional status (KPS), and inflammation (albumin, C-reactive protein) [21].

### 2.4. Access Route for Artificial Nutrition

The European Society for Clinical Nutrition and Metabolism (ESPEN) Guidelines were applied to choose the most appropriate access route for artificial nutrition [22,23]. The primary option for patients with adequate intestinal function was home enteral nutrition (HEN) via tube feeding. The preferred choice for patients with insufficient intestinal function was the home parenteral nutrition (HPN) via central venous catheter [12,22].

The nutrition-dedicated nurse trained the patient and the caregiver at home to manage the HAN independently. The training for HEN lasted about 1–3 days. For the HPN, the nurse trained the caregiver to follow the correct asepsis rules for infusion line management and early detection of complications [23]. This specialized training made the caregiver able to manage independently and safely the infusion line allowing the personalized self-administration of the HAN, according to the need and the preferences of the patient and reducing the risk of the main complications related to HAN, such as catheter-related bloodstream infection. The nurse always dressed the HAN access routes 1–2 times/week. HAN was regularly monitored: 1–2 times/week by the nutritionist and several times/week (as needed) by the nurse. Clinical, nutritional, and biochemical data were recorded in a nutrition repository during follow-up visits. For any emergency associated to HAN, the patient and/or the caregiver could call the nutritionist or nurse.

The National Health System provided all the material (blends and infusion sets for the HEN, bags containing standard formulas and material for attaching-detaching the HPN, and material for the dressing of the access routes) for the HAN. The ANT Foundation’s Family Service delivered material for the HAN to the patient’s home once a week.

### 2.5. Daily Intake and Expenditure

Daily energy (kcal) and protein (g) intake supplied by HAN were defined on the basis of energy requirement. Resting energy expenditure (REE, kcal) was calculated using the Harris–Benedict equation [24]; total energy expenditure (TEE, kcal) was estimated applying a correction activity factor to REE based on the KPS (1.1 if KPS is 40 or lower; 1.2 if KPS is 50; and 1.3 if KPS is 60 or higher) [25].

A nutritional investigation of the composition and frequency of meals estimated the energy and protein intake in patients preserving a partial oral food intake. HAN energy balance (kcal) was calculated as the difference between calories supplied by HAN and TEE; total energy balance was calculated as the difference between HAN + oral intake and TEE (kcal).

### 2.6. Nutritional and Clinical Outcomes

Weight variation was calculated as the difference between weight after 1 month and weight at the start of HAN; KPS variation was calculated as the difference between KPS after 1 month and KPS at the start of HAN.

The survival of patients (days) was evaluated from the start of HAN.

### 2.7. Statistical Analysis

Baseline characteristics of the patients entering in HAN were described as number (percentages) for categorical variables (gender, tumor primary site, cancer diffusion, stage of cachexia) and as mean ± standard deviation (SD) for quantitative variables (age, KPS and body mass index, BMI).

The comparison between patients surviving 6 weeks or less and surviving more than 6 weeks was analyzed by Mann–Whitney U test (age, KPS, and BMI at the entry) and chi-square test (gender, tumor primary site, cancer diffusion, and stage of cachexia).

The correlation between total energy balance and weight and KPS variation after 1 month was analyzed by Spearman rank correlation.

The association between weight and KPS variation after 1 month (dependent variables) and patient condition at the entry (KPS, BMI, cachexia, independent variables) was analyzed by a general linear model adjusted for age and gender. Patients with pre-cachexia are considered as a reference for the regression analysis.

The association between mortality and cachexia degree at the entry (KPS, BMI, cachexia) was analyzed by a Cox regression adjusted for age, gender, KPS, and BMI at the entry. Patients with pre-cachexia are considered as a reference for the regression analysis.

The differences between patients with no or local metastasis and with widespread metastasis were analyzed by chi-square test (for cachexia), Mann–Whitney U test (BMI and KPS at the entry, weight and KPS variation after 1 month, energy expenditure, and daily intakes) and log-rank Mantel–Cox test (survival).

All the analyses were executed using SPSS version 27 (IBM, Armonk, NY, USA).

## 3. Results

### 3.1. Clinical and Nutritional Features

Data were collected from 1 July 1990 to 30 June 2021, by the NST of the ANT Foundation operating in Bologna and its province in Italy. Within the 45,382 cancer patients assisted at home by ANT during this period, HAN was planned in 1063 patients (2.3% of total, 352 HEN and 711 HPN). The overall median survival for the whole cohort of patients starting HAN was 87 days (95% C.I. 81–93 days). Among them, 101 suspended artificial nutrition for oral refeeding and 962 continued HAN until death, of which 226 (23.5%) survived 6 weeks or less (median survival 28 days, 95% C.I. 26–30 days) and 736 (76.5%) survived more than 6 weeks (median survival 103 days, 95% C.I. 96–110 days) (Table 2).

At the nutritionist’s first visit, all patients had a negative energy balance with progressive weight loss. Among them, 752 patients (70.7%) did not feed at all, and the remaining 311 patients (29.3%) had a low mean daily oral energy intake (280 ± 69 kcal). Malnourished patients were only 447 (42.0%). 

The primary tumor site was the gastrointestinal tract (54.7%) and head/neck region (25.6%). Cancer was in an advanced stage with widespread metastases in 59.2% of patients starting HAN, up to 83.2% in those who survived 6 weeks or less, and significantly more than 48.4% in patients surviving more than 6 weeks (*p* < 0.001). 

At the start of HAN, patients surviving more than 6 weeks had a higher KPS and a lower degree of cachexia when compared to patients surviving less than 6 weeks (*p* < 0.001); BMI was similar. 

The complications of HAN involved 142 out of 1063 (13.3%) patients. The most frequent complication for the enteral tube were occlusion (21 patients) and spontaneous removal (13 patients). Catheter-related bloodstream infection (CRBSI, 35 patients) and catheter-related thrombosis (13 patients) were the major complications of HPN. CRBSI required hospitalization for 17 patients.

### 3.2. Nutritional and Clinical Outcomes

The analysis of nutritional and clinical outcomes was performed on the 736 patients surviving more than 6 weeks, since in these patients the measuring of weight and KPS after 1 month from the start of HAN was considered evaluable and reliable. The worsening of clinical condition consented the evaluation of weight and KPS variation only for 72 of 226 patients surviving 6 weeks or less (31.8%); these patients showed a decrease in body weight (−0.1 ± 0.6 kg) and in KPS (−1.4 ± 4.6) after 1 month.

Total energy balance, evaluated on all the patients surviving more than 6 weeks, was positive (+281 ± 380 kcal/day) and correlated with the improvement of body weight (+0.3 ± 0.7 kg) and KPS (+2.3 ± 5.7) after 1 month of HAN (Figure 1).

Table 3 shows the association between weight and KPS variation and conditions of patients at the start of HAN. Patients with cachexia and refractory cachexia showed less improved weight and KPS after 1 month of HAN (*p* < 0.001). Higher BMI at the entry was negatively associated with weight and KPS improvement (*p* < 0.001).

The association between mortality and cachexia degree at the entry is shown in Figure 2. Patients starting HAN in cachexia or refractory cachexia had an increased mortality hazard (odds ratio 2.165 and 3.590, respectively, *p* < 0.001), which referred to patients in pre-cachexia. Patients with refractory cachexia showed a shorter survival (median survival 83 days, 95% C.I. 139–175 days) than patients with cachexia (median survival 97 days, 95% C.I. 90–104 days) or pre-cachexia (median survival 157 days, 95% C.I. 137–175 days; *p* < 0.001).

### 3.3. No-Local vs. Widespread Metastasis

The advanced stage of cancer greatly affected the possibility of improving nutritional and performance status (Table 4). Patients with widespread metastasis had an advanced degree of cachexia (*p* = 0.011) and less survival (*p* < 0.001).

REE and TEE were significantly lower in patients with widespread metastasis than in patients with localized cancer. Despite that HAN and total energy intake were similar or mildly greater in patients with widespread metastasis, patients with advanced cancer improved weight significantly less (*p* = 0.014) than patients with localized cancer.

Patients with widespread metastasis had a lower KPS at the entry (*p* < 0.001) and a poorer KPS improvement after 1 month of HAN (*p* = 0.022) compared to patients with no or local metastasis.

## 4. Discussion

In July 1990, the ANT Foundation launched a nutrition counseling service in Bologna (Italy) as part of its home palliative care program [11]. The ANT Foundation, founded in 1978, is a non-profit organization providing free home care to cancer patients [15,26]. Since 1985, the ANT Foundation has assisted 130,000 cancer patients in 11 (out of 20) Italian regions through 23 multidisciplinary teams, being the widest experience in free home care for cancer patients in Italy and Europe.

Artificial nutrition may be used in home palliative care when an improvement of the quality of life is expected, and the probability of death from malnutrition is greater than the risk deriving from cancer progression [1,12,27]. HAN enables the patients to live the last months in their familiar environment, and is significantly less expensive for the National Health System with respect to artificial nutrition carried out in a hospital, as reported in our previous articles [11,18]. Nevertheless, the decision to initiate HAN in patients with advanced cancer cannot be based solely on the presence of malnutrition, but should take into account bioethical considerations such as the appropriateness to feed patients who are expected to live for only a few weeks or days [28,29,30]. Therefore, the use of valid selection criteria for HAN eligibility plays a critical role in identifying patients who can benefit from HAN and reduces the risk of an excessive and unrestrained nutritional therapy, which could result in therapeutic obstinacy.

One of the most important criteria for eligibility to HAN has been the estimation of survival. Whereas death from loss of lean body mass occurs after approximately 60–75 days of starvation in healthy subjects [31], in patients with advanced cancer, prolonged fasting combined with underlying cachexia and increased nitrogen catabolism diminishes survival to about 35–40 days. Therefore, 6 weeks may be considered an acceptable life expectancy for starting HAN [32]. Our study showed that in patients surviving 6 weeks or less, HAN was not able to prevent weight loss and KPS impairment. These findings suggest that HAN did not avoid death from cachexia, resulting in it being probably useless, expensive, and worsening the quality of life in patients with a prognosis lower than 6 weeks. 

The survival of patients selected for HAN in our home palliative “real world” setting was more than 6 weeks in 76.5% of cases. Although encouraging, these data indicate that approximately one-quarter of the patients died in six weeks or less. This could be due to the unpredictability of the cancer disease trend as well as to the not absolute reliability of the Palliative Prognostic Score in predicting life expectancy, but often the choice of satisfying the desire of the patient at the end of life to be nourished, even artificially, has been influenced by ethical consideration. 

Cancer cachexia, according to international consensus conferences, is a multifactorial syndrome characterized by a continuous loss of skeletal muscle mass, supported or not by a loss of fat mass. Cancer cachexia produces outcomes in a gradual functional impairment and cannot be fully recovered by standard nutritional support [21,33]. This definition distinguishes cancer cachexia from simple starvation or age-related loss of muscle mass. Since cancer cachexia causes a weight loss with a multifactorial origin, a proper nutritional therapy, such as artificial nutrition, cannot always recover nutritional and functional status [20]. Our results emphasize the importance of the cachexia status evaluation before initiating HAN. Weight and KPS variations were greater in patients entering with pre-cachexia, whereas patients initiating HAN with cachexia or refractory cachexia had a poorer survival and a reduced response to the favorable effects of artificial nutrition. This result confirms the scarce benefit of HAN in advanced cancer patients with refractory cachexia. For them, the decision to begin the HAN should therefore be based mostly on ethical considerations rather than clinical/nutritional parameters.

Previous research examined the impact of caloric intake and energy expenditure on the nutritional status of patients at diverse stages of cancer disease, founding a significant correlation between weight loss, performance status, and survival [34,35,36,37,38]. Malnutrition reduces muscle strength and a patient’s autonomy in everyday activities, resulting in fatigue, malaise, and depression. In addition, malnutrition impairs immune function, affects the responses to chemotherapy, and enhances chemotherapy-induced toxicity and complications. Clinicians may overlook or undertreat the risk of cancer-related malnutrition [39,40]. When the reduction in oral food intake is attributable to organic consequences of cancer or treatment-related adverse events, artificial nutrition may be an adequate nutrition therapy for cancer cachexia [11].

The stage and diffusion of cancer greatly affect the energy requirements of oncological patients. Hypermetabolism, more than 10% of REE, is characterized by an increase in the body’s basal metabolic rate, and is present in more than 50% of cancer patients [41]. The energy requirements of the neoplastic mass, the development of a chronic systemic inflammatory state in the host, and the production of lipid mobilizing factors and proteolysis inducing factors by cancer itself [21,42] increase the energy and protein catabolism and cause a higher degree of malnutrition than that due only to the reduction of oral food intake, compromising the expected response to the artificial nutrition [43].

In this context, the correct assessment of energy requirement becomes a crucial step in choosing the type and composition of the nutritional therapy. Several studies [42,43,44,45,46] emphasize the importance of indirect calorimetry as the goal standard in evaluating the real energy requirements in cancer patients. Predictive formulas, such as the Harris–Benedict equation, do not consider the cancer energy expenditure, underestimating the real REE of the patient [47].

The present retrospective observational study confirms this hypothesis. Although the HAN made positive energy balance improving weight, performance status, and survival, patients with advanced cancer and widespread metastases improved weight and KPS less than patients with localized cancer, even if they received an equivalent HAN protein-calorie intake. These patients had a high degree of cachexia at the entry and the REE, assessed with the Harris–Benedict predictive formula, was lower than patients with localized cancer. In these patients, the energy expenditure was probably underestimated, therefore the use of an appropriate instrument taking into account cancer energy requirements, such as the indirect calorimetry, would optimize the HAN protein-calorie prescription. 

This study lacks a validated questionnaire-based assessment of the quality of life. Evidently, quality of life questionnaires are burdensome for these patients and do not collect the potential changes induced by HAN. A better determination of the cachexia degree would necessitate the analysis of body composition, which was not performed in all patients but only recently becomes part of our investigative background. Therefore, we could not evaluate if body weight changes were due to variation of hydration status and/or fat loss or gain.

Resting energy expenditure was calculated exclusively using a predictive formula. A better determination of the cancer patient’s metabolism requires an indirect calorimetry, which, however, is often inconvenient and cumbersome for the patient and very expensive. The recent acquisition of a handheld device for the analysis of indirect calorimetry will allow us, in future studies, to better evaluate the patient’s energy balance and respond more adequately to the patient’s requirements with artificial nutrition. Finally, the total energy expenditure was not directly measured but estimated by applying correction factors to the Harris–Benedict formula. The correction factors were evaluated according to the level of physical activity supposed on the basis of KPS. 

## 5. Conclusions

The positive energy balance provided by HAN improves the nutritional and performance status in cancer patients with a life-expectancy longer than 6 weeks. Advanced cancer and refractory cachexia negatively affect the positive impact of HAN.

## Figures and Tables

**Figure 1 nutrients-14-04307-f001:**
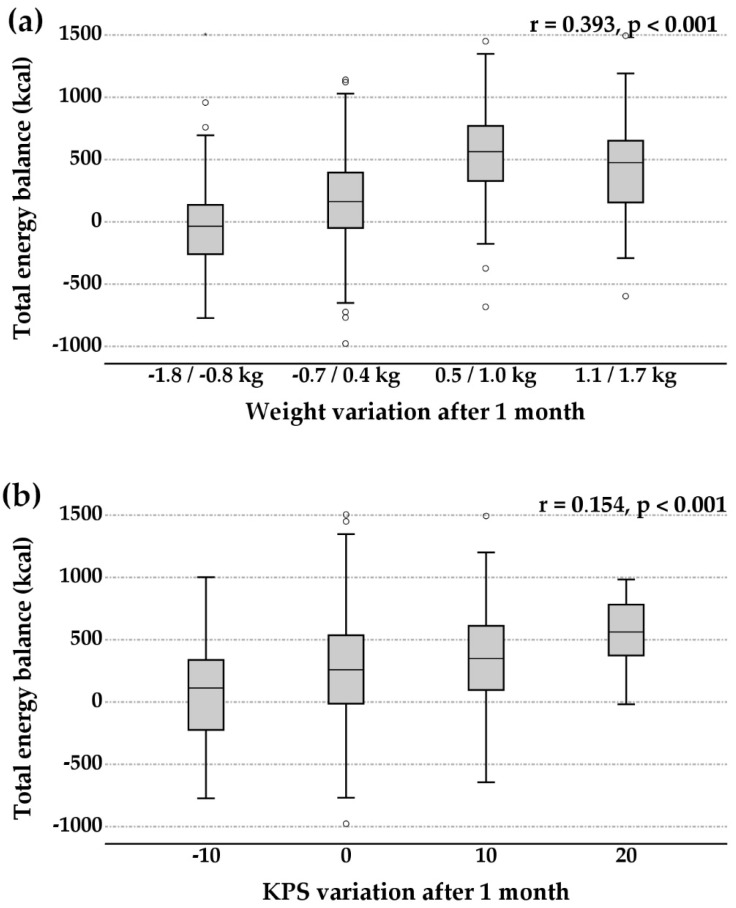
Weight and KPS variation after 1 month according to the total energy balance. Statistical analysis was performed by Spearman rank correlation.

**Figure 2 nutrients-14-04307-f002:**
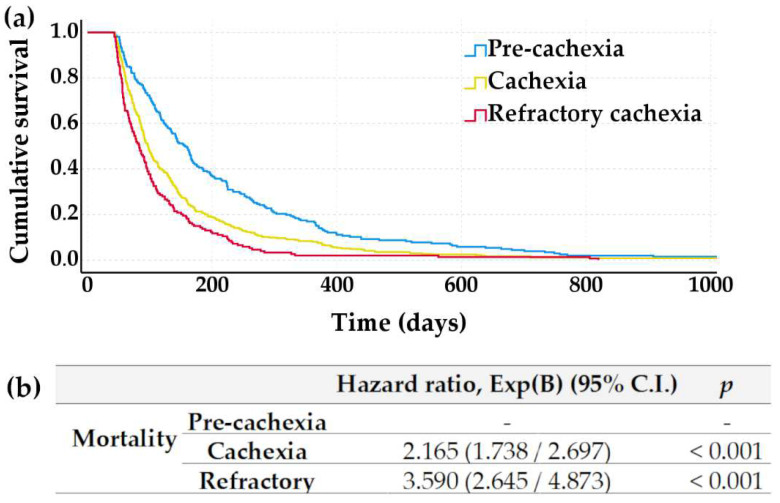
Survival according to cachexia stage at the entry in HAN for patients surviving more than 6 weeks. (**a**) Kaplan–Meier survival curves for patients in pre-cachexia, cachexia, and refractory cachexia; (**b**) Association between mortality and cachexia degree at the entry (KPS, BMI, cachexia) was analysed by a Cox regression adjusted for age, gender, KPS, and BMI at the entry. Patients with pre-cachexia are considered as a reference for the regression analysis.

**Table 1 nutrients-14-04307-t001:** Steps to define HAN eligibility.

** *(1). Negative energy balance ± malnutrition* **	Negative energy balance: oral intake lower than 50% of energy requirements for more than 1–2 weeks, with consequent weight loss.
	Protein-calorie malnutrition: BMI < 18.5 kg/m^2^ and weight loss in the last 6 months >10%.
** *(2). Life expectancy* **	Estimated life expectancy (KPS and Palliative Prognostic Score) ≥ 6 weeks.
** *(3). Physical, psychological and environmental conditions* **	No severe organ failure.
	Well controlled pain.
	Patient and/or caregiver able to manage independently HAN.
	Adequate environmental and hygiene conditions.
** *(4). Informed consent* **	Written informed consent for treatment and data collection from the patient and caregiver.

**Table 2 nutrients-14-04307-t002:** Baseline characteristics of the patients starting HAN.

		Patients Continued HAN Until Death		*p* ^#^
	Patients Starting HAN (*n* = 1063)	Surviving 6 Weeks or Less (*n* = 226)	Surviving More than 6 Weeks (*n* = 736)	
**Age**, mean ± SD	65.6 ± 12.7	65.4 ± 11.7	66.3 ± 12.9	0.209
**Gender**, n (%)				
Men	618 (58.1)	139 (61.5)	419 (56.9)	0.248
Women	445 (41.9)	87 (38.5)	317 (43.1)	
**Tumour primary site**, n (%)				
Gastrointestinal	581 (54.7)	122 (54.0)	432 (58.7)	<0.001
Head and neck	272 (25.6)	34 (15.0)	172 (23.4)	
Genital	85 (8.0)	23 (10.2)	59 (8.0)	
Respiratory	48 (4.5)	23 (10.2)	24 (3.3)	
Other *	71 (7.2)	24 (10.6)	49 (6.6)	
**Cancer diffusion**, n (%)				
No metastasis	238 (22.4)	4 (1.8)	155 (21.1)	<0.001
Local metastasis	196 (18.4)	34 (15.0)	151 (20.5)	
Widespread metastasis	629 (59.2)	188 (83.2)	430 (58.4)	
**KPS**, mean ± SD	51.7 ± 9.9	47.3 ± 7.4	52.1 ± 9.6	<0.001
**Stage of cachexia**, n (%)				
Pre-cachexia	289 (27.2)	17 (7.5)	207 (28.1)	<0.001
Cachexia	518 (48.7)	111 (49.1)	375 (51.0)	
Refractory cachexia	256 (24.1)	98 (43.4)	154 (20.9)	
**BMI** (kg/m^2^), mean ± SD	19.5 ± 3.1	19.2 ± 2.9	19.4 ± 3.0	0.194

HAN, home artificial nutrition; KPS, Karnofsky Performance Status; BMI, body mass index. * Breast, urinary, haematological, nervous system, bone, and soft tissues. ^#^
*p* referred to the comparison between patients surviving less than 6 weeks and patients surviving more than 6 weeks. Statistical analysis was performed by Mann–Whitney U Test for age, KPS, and BMI and by chi-square test for gender, tumor primary site, cancer diffusion, and stage of cachexia.

**Table 3 nutrients-14-04307-t003:** Association between weight and KPS variation after 1 month (dependent variables) and patient condition at the entry (KPS, BMI, cachexia, independent variables) in patients surviving more than 6 weeks. The analysis was performed by a general linear model adjusted for age and gender. Patients with pre-cachexia are considered as a reference for the regression analysis.

Dependent Variable	Indipendent Variables	Standardized β Coefficient (95% C.I.)	*p*
**Weight variation** (adjusted R^2^ = 0.126)	KPS	0.000 (−0.005/0.005)	0.927
	BMI	−0.110 (−0.133/−0.087)	<0.001
	Pre-cachexia	-	-
	Cachexia	−0.349 (−0.488/−0.210)	<0.001
	Refractory cachexia	−0.614 (−0.809/−0.419)	<0.001
**KPS variation** (adjusted R^2^ = 0.120)	KPS	−0.157 (−0.198/−0.115)	<0.001
	BMI	−0.488 (−0.674/−0.301)	<0.001
	Pre-cachexia	-	-
	Cachexia	−2.924 (−4.050/−1.798)	<0.001
	Refractory cachexia	−5.713 (−7.294/−4.132)	<0.001

**Table 4 nutrients-14-04307-t004:** Nutritional outcome, survival, and protein-calorie intake of patients with no or local metastasis and with widespread metastasis.

	No or Local Metastasis (*n* = 306)	Widespread Metastasis (*n* = 430)	*p*
Stage of cachexia, *n* (%)			0.011
Pre-cachexia	98 (32.0)	109 (25.3)	
Cachexia	159 (52.0)	216 (50.2)	
Refractory cachexia	49 (16.0)	105 (24.5)	
BMI at the entry, mean ± SD	19.4 ± 2.9	19.4 ± 3.0	0.591
Weight variation (kg), mean ± SD	0.4 ± 0.7	0.3 ± 0.7	0.014
KPS at the entry, mean ± SD	53.8 ± 9.6	51.0 ± 9.5	<0.001
KPS variation, mean ± SD	2.7 ± 5.7	2.1 ± 5.7	0.022
Survival, median (95% C.I.)	144.0 (131.9–156.1)	86.0 (80.4–91.6)	<0.001
REE (kcal)	1201 ± 173	1168 ± 187	0.008
TEE (kcal)	1470 ± 241	1403 ± 255	<0.001
HAN energy intake (kcal)	1644 ± 415	1621 ± 360	0.279
HAN energy intake/body weight (kcal/kg)	30.9 ± 9.0	31.6 ± 8.5	0.275
HAN energy balance (kcal)	+173 ± 416	+218 ± 366	0.133
HAN protein intake/body weight (g/kg)	1.2 ± 0.4	1.3 ± 0.4	0.035
Total energy intake (kcal)	1722 ± 415	1706 ± 370	0.374
Total energy intake/body weight (kcal/kg)	32.4 ± 9.3	33.3 ± 8.7	0.207
Total energy balance (kcal)	+251 ± 413	+302.1 ± 373.0	0.085
Total protein intake/body weight (g/kg)	1.3 ± 0.4	1.4 ± 0.4	0.041

REE, resting energy expenditure; TEE, total energy expenditure; HAN, home artificial nutrition; HAN energy balance is calculated as HAN energy intake—TEE; Total energy balance is calculated as total energy intake—TEE. Statistical analysis was performed by chi-square test (for cachexia), Mann–Whitney U test (for quantitative variables), and log-rank Mantel–Cox (survival).

## Data Availability

Not applicable.

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
