# Peer review of "Home Artificial Nutrition and Energy Balance in Cancer Patients: Nutritional and Clinical Outcomes"

_nutrients, 2022, doi:10.3390/nu14204307_

Round 1

Reviewer 1 Report

Thank you for an opportunity to review the manuscript titled: “Home Artificial Nutrition and Energy Balance in Cancer Patients: Nutritional and Clinical Outcomes”.

This manuscript investigates important topic of home artificial nutrition in home palliative care setting for cancer patients. It is well written and well presented.

Comments/suggestions to improve the manuscript:

Introduction:

-          Expand on the reasons why consideration of quality of life is essential in this patient cohort to highlight this issue to the readers.

Methods:

-          Could you explain why one of the eligibility criteria was for “Patient and/or caregiver able to manage independently HAN”? Is it due to a lack of homecare nursing capacity? This is interesting as in other European settings, e.g. in the UK, the majority of the HAN administration (especially HPN) is undertaken by the homecare nursing staff.

-          Could you clarify if the establishment of the HAN was undertaken at home or in the hospital?

Results:

-          Could you provide an overall median survival for the whole cohort to facilitate comparison between the studies?

-          Could you clarify when was the first nutritionist visit?

-          It would be interesting to see whether there were any differences in survival as well as baseline characteristics between patients initiated on HEN versus HPN.

-          Could you present the outcomes (weight variation/KPS variation) for longer period of time also e.g. 2 months/ 3months/ 6 months?

-          Did you collect any data on complications associated with HAN e.g. catheter-related blood stream infections / CVC thrombosis or data about hospital readmissions/ place of death?

Conclusions:

-          As the energy balance/KPS were not improved in the patient cohort who did not survive more than 6 weeks I would recommend revision of the sentence “The positive energy balance provided by HAN improves the nutritional and performance status in cancer patients.”

Minor points:

-          Consider merging first four paragraphs of introduction into one to improve the flow of reading.

-          Limitations could be moved to the last paragraph of discussion.

Author Response

Comments and Suggestions for Authors

Thank you for an opportunity to review the manuscript titled: “Home Artificial Nutrition and Energy Balance in Cancer Patients: Nutritional and Clinical Outcomes”.

This manuscript investigates important topic of home artificial nutrition in home palliative care setting for cancer patients. It is well written and well presented.

Thank you very much for your positive feedback.

Comments/suggestions to improve the manuscript:

Introduction:

-          Expand on the reasons why consideration of quality of life is essential in this patient cohort to highlight this issue to the readers.

Thank you for this suggestion, we explained better this point and we added two references in the introduction (lines 41-48).

Methods:

-          Could you explain why one of the eligibility criteria was for “Patient and/or caregiver able to manage independently HAN”? Is it due to a lack of homecare nursing capacity? This is interesting as in other European settings, e.g. in the UK, the majority of the HAN administration (especially HPN) is undertaken by the homecare nursing staff.

We explained that a specific training by a specialized nurse made the caregiver able to manage independently and safely the infusion line allowing the personalized self-administration of the HAN, according to the need and the preferences of the patient, and reducing the risk of the main complications related to HAN, such as catheter-related bloodstream infection (lines 104-108). This approach resulted more effective than a continuous and constant intervention of a not-dedicated (and not-specialized) nurse.

-          Could you clarify if the establishment of the HAN was undertaken at home or in the hospital?

We clarified this point (lines 79-82). In particular, in most cases, artificial nutrition was set up at home (62%), in a third of patients (38%) the artificial nutrition was set up in the hospital.

Results:

-          Could you provide an overall median survival for the whole cohort to facilitate comparison between the studies?

The overall median survival was 87 days, we added this data (lines 159-160).

-          Could you clarify when was the first nutritionist visit?

The nutritionist first visit was required by the home palliative care physician when, after the evaluation of the nutritional status, considered the intervention of the nutritional service team to be necessary (lines 67-70).

-          It would be interesting to see whether there were any differences in survival as well as baseline characteristics between patients initiated on HEN versus HPN.

Thank you, this is a very interesting point. In this study, we did not discuss the differences between patients in HEN and HPN because they were extensively treated in our previous paper (Ruggeri et al., 2020, Clinical Nutrition, doi:10.1016/j.clnu.2020.02.021).

-          Could you present the outcomes (weight variation/KPS variation) for longer period of time also e.g. 2 months/ 3months/ 6 months?

The outcomes for a longer period were not evaluated. The evaluation of one-month outcomes aimed to describe the short-term impact of HAN in a cohort of palliative care patients with advanced cancer (poor life expectancy and progressive and rapid decline of clinical condition).

 -          Did you collect any data on complications associated with HAN e.g. catheter-related blood stream infections / CVC thrombosis or data about hospital readmissions/ place of death?

Thank you for the suggestion, we added this point in the result session (lines 175-179). For the aim of this paper, we considered not indispensable to discuss the differences between HEN and HPN as well as their complications because we reported the details of these aspects in our previous paper (Ruggeri et al., 2021 doi: 10.1016/j.nut.2021.111264).

Conclusions:

-          As the energy balance/KPS were not improved in the patient cohort who did not survive more than 6 weeks I would recommend revision of the sentence “The positive energy balance provided by HAN improves the nutritional and performance status in cancer patients.”

Thank you, we revised the sentence (lines 337-338).

Minor points:

-          Consider merging first four paragraphs of introduction into one to improve the flow of reading.

Ok, done.

-          Limitations could be moved to the last paragraph of discussion.

Ok, done.

Reviewer 2 Report

 Home Artificial Nutrition and Energy Balance in Cancer Patients: Nutritional and Clinical Outcomes

His is interesting article presenting long time observation period.

My remarks:

·       Definition of pre-cachexia and refractor cachexia was introduced b Kenneth Fearon et all in 2011. (ref 20 in our aritcle). How is it possible that in 1990 during qualifiaction before start of HAN investigators assesed this criterium.

·       Could you please explain diagnostic criteria of malnutrition, that you used. If patient does not eat, he is malnourished, so maybe 70% were malnourished? Or maybe my English is not fluent enough.

·       „At the nutritionist's first visit, all patients had a negative energy balance with progressive weight loss, and 447 (42.0%) were malnourished. Most of the patients (n=752, 70.7%) did not feed at all.”  

·       In article that y ou cited, I couldn’t find indicators for Karnoffsky  score, that you used in this article.

·       Another thing, in discussion please comment on body  composition. It’s widely  known, that body  weight gain in cancer patient is mostly  due to water and fat gain. It was week aspect of your trial and you should mention it.

Author Response

Comments and Suggestions for Authors

Home Artificial Nutrition and Energy Balance in Cancer Patients: Nutritional and Clinical Outcomes

His is interesting article presenting long time observation period.

Thank you very much for your positive feedback.

My remarks:

  • Definition of pre-cachexia and refractor cachexia was introduced b Kenneth Fearon et all in 2011. (ref 20 in our aritcle). How is it possible that in 1990 during qualifiaction before start of HAN investigators assesed this criterium.

Thank you for this important comment. We used data collected from 1990 (weight loss, BMI, anorexia, sarcopenia, performance status, inflammation, etc…) to classify all the patients according Fearon’s definition of pre-cachexia, cachexia and refractory cachexia.

  • Could you please explain diagnostic criteria of malnutrition, that you used. If patient does not eat, he is malnourished, so maybe 70% were malnourished? Or maybe my English is not fluent enough.

 „At the nutritionist's first visit, all patients had a negative energy balance with progressive weight loss, and 447 (42.0%) were malnourished. Most of the patients (n=752, 70.7%) did not feed at all.”  

Thank you for this observation. The criteria to define malnutrition are reported in table1 (BMI < 18.5 kg/m2 and weight loss in the last 6 months > 10%), but the principal step to define HAN eligibility was the negative energy balance (table 1, oral intake lower than 50% of energy requirements for more than 1-2 weeks), with or without malnutrition. We modified table 1 to highlight this aspect. Moreover, we slightly changed the results (lines 164-167) to make clear that all the enrolled patients had a negative energy balance. Among them, 752 patients (70%) did not feed at all (but they were not necessarily malnourished, according to the previous parameters).

  • In article that you cited, I couldn’t find indicators for Karnoffsky score, that you used in this article.

Not having the possibility to measure the total energy expenditure, we estimated it applying correction factors to Harris Benedict formula. In our opinion, the most appropriate parameter to evaluate the activity level of our patients was the KPS thus we took the article by Ceolin Alves et al. as a model to get the correction factors. We added this important point to limitations (lines 333-335).

  • Another thing, in discussion please comment on body  composition. It’s widely  known, that body  weight gain in cancer patient is mostly  due to water and fat gain. It was week aspect of your trial and you should mention it.

Thank you for this suggestion, we added this point to limitations (lines 324-326).

Reviewer 3 Report

Thsnk you for your nice and interesting article. 

It's an important issue, we need to discuss who, how and when to feed patients with cancer and cachexia. 

Line 10 -11 in the abstract is not clear. 

I would prefer sub-titles in the abstract. 

There are few passages which need to be re-written. 

Overall, an informative and important paper. 

Author Response

Recommendation: Accept after minor revision (corrections to minor methodological errors and text editing)

Comments and Suggestions for Authors:

Thank you for your nice and interesting article.

It's an important issue, we need to discuss who, how and when to feed patients with cancer and cachexia.

- Line 10 -11 in the abstract is not clear.

We modified the abstract in order to clarify the point you suggested.

- I would prefer sub-titles in the abstract.

Thank you, we agree with you but we wrote the abstract according to Nutrients guidelines.

- There are few passages which need to be re-written.

Thank you, we rephrased some sentences.

Overall, an informative and important paper.

Thank you very much for your positive feedback.